# Safety Profile of the 4CMenB (Bexsero^®^) Vaccine: A Systematic Review and Meta-Analysis of Adverse Events in Clinical Trials

**DOI:** 10.3390/vaccines13101030

**Published:** 2025-10-02

**Authors:** Ana Belén García Flores, Rafael Ruiz-Montero, María Ángeles Onieva-García, Alexander Batista-Duharte, Estefanía López Cabrera, Mohamed Farouk Allam, Inmaculada Salcedo

**Affiliations:** 1Preventive Medicine and Public Health Unit, Maimonides Biomedical Research Institute of Cordoba (IMIBIC), Reina Sofia University Hospital, 14004 Cordoba, Spain; h82gafla@uco.es (A.B.G.F.); rafael.ruiz@uco.es (R.R.-M.); h42ongam@uco.es (M.Á.O.-G.); estefania.lopez.cabrera.sspa@juntadeandalucia.es (E.L.C.); fm2faahm@uco.es (M.F.A.); 2Preventive Medicine and Public Health, Department of Medical and Surgical Sciences, University of Cordoba, 14004 Cordoba, Spain; 3Department of Cell Biology, Physiology and Immunology, University of Cordoba, 14004 Cordoba, Spain; 4Immunology and Allergy Group (GC01), Maimonides Biomedical Research Institute of Cordoba (IMIBIC), Reina Sofia University Hospital, 14004 Cordoba, Spain

**Keywords:** Bexsero^®^, 4CMenB, adverse events, clinical trials, meta-analysis

## Abstract

Background: The 4CMenB vaccine (Bexsero^®^) contains surface proteins from *Neisseria meningitidis* serogroup B and is recommended from 2 months of age. The most frequently reported adverse events are fever, injection site pain, and fatigue. Thus, this study aimed to estimate the incidence of local and systemic adverse events associated with the administration of the 4CMenB (Bexsero^®^) vaccine. Methods: A systematic review and meta-analysis of clinical trials published up to 28 February 2025 were conducted using PubMed, ScienceDirect, and Web of Science. Human studies available in English, Spanish, French, German, or Italian were exclusively included. Adverse events following the first dose of the vaccine were analyzed. Pooled proportions with 95% confidence intervals were calculated, and heterogeneity across studies was assessed using the I^2^ statistics. Results: Ten clinical trials comprising 13,345 participants were included. The most common adverse event was local pain (occurring in up to 94% of cases), followed by induration, erythema, and edema, with frequencies ranging from 25% to 45%. The most frequently reported systemic events were irritability (up to 75%), fatigue (51–59%), fever (up to 60%), headache (42–49%), and persistent crying (50–65%). Most adverse events were mild and self-limiting. Conclusions: The 4CMenB (Bexsero) vaccine exhibits a favorable safety profile, characterized by a predominance of mild and transient local adverse events. Although several systemic events were reported, their overall frequency was generally low. These findings support the continued inclusion of Bexsero^®^ in routine childhood immunization programs.

## 1. Introduction

Meningococcal disease (MD) remains a major public health concern worldwide. Invasive meningococcal disease (IMD), which is associated with high morbidity and mortality, is predominantly caused by five serogroups of *Neisseria meningitidis*—A, B, C, W, and Y [1]. Approximately 5–10% of patients with IMD die within the first 48 h after symptom onset, while 10–20% of survivors experience serious sequelae. Reported long-term complications include sensorineural hearing loss, neurodevelopmental impairment, limb amputations, and cognitive deficits, with the overall case fatality rate ranging between 10% and 15% [2,3].

Currently, serogroup B of *N. meningitidis* is the leading cause of IMD in Europe and North America and remains among the most prevalent serogroups in Latin America [4,5]. Risk factors associated with MD acquisition and progression to invasive disease include genetic susceptibility, concurrent respiratory infections, tobacco smoke exposure, and living conditions characterized by overcrowding and low socioeconomic status [6,7]. The bacterium colonizes the human nasopharynx, with asymptomatic carriage rates estimated at 10–20% of the population. Transmission occurs through respiratory droplets, and disease development depends on the pathogen’s ability to cross the epithelial barrier and invade the bloodstream [8].

Due to its rapid onset, high mortality, and substantial long-term disability burden, prevention of MD remains a major challenge for global health authorities. Although advances in clinical management have helped reduce mortality, vaccination remains the most effective strategy against invasive forms and their severe consequences [9].

In Spain, meningococcal vaccination began in 2000 with the introduction of the conjugate vaccine against serogroup C, administered at 4 months, 12 months, and 12 years of age according to the national schedule. In 2019, the Public Health Commission approved the inclusion of the quadrivalent ACWY vaccine [10]. Since September 2015, the serogroup B meningococcal vaccine (Bexsero^®^, 4CMenB) has also been available. Although serogroup B was the predominant cause of IMD, its incorporation into the national immunization program was initially delayed due to cost-effectiveness considerations [11]. In June 2018, the European Medicines Agency (EMA) approved a 2 + 1 schedule to begin vaccination between 3 and 5 months of age, following the model used in the United Kingdom. Full vaccination with 4CMenB has demonstrated 71% effectiveness in preventing invasive disease caused by both serogroup B and non-B meningococci in children under 5 years of age [12].

In Andalusia, the #ZeroMeningitis campaign was launched in January 2020 to minimize disease burden and complications. Initial measures included replacing the serogroup C vaccine with MenACWY at 12 months and 12 years, as well as a catch-up campaign for adolescents aged 13–18 years between 2020 and 2022. In December 2021, the 4CMenB vaccine was incorporated into the regional schedule for all infants born after 1 October 2021, using the 2 + 1 scheme (2, 4, and 15 months) [13].

4CMenB has been associated with higher rates of fever (>38 °C) and local reactions compared to routine childhood vaccines, with these rates increasing further when co-administered with other immunizations. For this reason, separate administration and the prophylactic use of paracetamol have been recommended [14,15,16]. However, despite its demonstrated effectiveness in preventing invasive meningococcal disease [4,17,18,19,20,21,22,23,24,25,26,27,28,29,30,31,32,33,34,35], concerns regarding reactogenicity remain a major barrier to broader acceptance and implementation in immunization programs. Given these considerations, the present study aims to assess the incidence of local and systemic adverse events following immunization with the 4CMenB (Bexsero^®^) vaccine in infants, children, and adults, compared with placebo or alternative vaccines.

## 2. Materials and Methods

### 2.1. Study Design

A systematic review and meta-analysis were conducted in accordance with the PRISMA 2020 guidelines (Preferred Reporting Items for Systematic Reviews and Meta-Analyses), aiming to quantitatively estimate the incidence of adverse events associated with the administration of the 4CMenB (Bexsero) vaccine. This review was internally registered on 8 May 2025, as part of the Bachelor’s Thesis of Ana Belén García Flores at the University of Córdoba, Spain.

### 2.2. Participants

Eligible participants were infants, children, or adults who received at least one dose of the 4CMenB vaccine for the first time, without restrictions on sex, race, or nationality.

### 2.3. Intervention

Vaccination with 4CMenB (Bexsero^®^), either as a single administration or co-administered with other routine vaccines.

### 2.4. Comparators

Although some trials included comparisons with placebo or other vaccines, the present meta-analysis focused exclusively on the frequency of adverse events after 4CMenB vaccination; therefore, direct group comparisons were not extracted.

### 2.5. Outcomes of Interest

All reported adverse events were included, both local (e.g., pain, erythema, induration, swelling at the injection site) and systemic (e.g., fever, fatigue, headache, irritability, vomiting).

### 2.6. Eligibility Criteria

Inclusion criteria comprised randomized phase II and III clinical trials conducted in humans in which 4CMenB was the primary intervention, published up to 28 February 2025, in English, Spanish, French, German, or Italian, and providing numerical data on the incidence of at least one adverse event. Exclusion criteria encompassed observational studies (cohort, case–control, or cross-sectional), systematic reviews, editorials, letters, opinion articles, preclinical or animal studies, and trials without extractable data or insufficient information to estimate proportions.

### 2.7. Information Sources and Search Strategy

A systematic search was performed in PubMed, ScienceDirect, and Web of Science from database inception until 28 February 2025. The following search strategy was applied using Boolean operators: (“Bexsero” OR “4CMenB”) AND (“adverse events” OR “adverse effects”) AND (“clinical trials”). Search strategies were adapted for each database when necessary. Reference lists of included studies were also screened to identify additional relevant articles. The search was limited to human studies and publications in the five predefined languages.

### 2.8. Study Selection

Records were managed with Microsoft Excel to remove duplicates. Two reviewers independently screened studies in two phases: (1) title and abstract review and (2) full-text evaluation. Discrepancies were resolved by consensus, and, when disagreement persisted, a third reviewer was consulted.

The systematic search (PRISMA Flow Diagram, Figure 1) initially identified 20 potentially relevant studies. After applying the inclusion and exclusion criteria, 10 studies were excluded because they did not provide quantifiable data on the frequency of adverse events related to the 4CMenB vaccine [21,22,23,24,25,26,27,28,29,30] (Appendix A).

### 2.9. Data Extraction

From each included study, we extracted the following information: authors and year of publication, country of study, total sample size, number of cases by type of adverse event (local or systemic), and the corresponding frequency with 95% confidence intervals. Data extraction was conducted independently by two reviewers, and any discrepancies were resolved through consensus or, when necessary, adjudication by a third reviewer.

### 2.10. Risk of Bias Assessment

A formal tool such as RoB 2 was not applied. However, since all included studies were randomized phase II or III trials published in peer-reviewed journals with adequate design and reporting, the overall risk of bias was judged to be low. This approach may be considered a limitation of the present review.

### 2.11. Data Synthesis and Statistical Analysis

Pooled proportions of adverse events were calculated as the number of reported cases divided by the total number of participants across the included studies. A random-effects model was used to account for expected heterogeneity, which was assessed using the chi-square (Q-test) and the I^2^ statistic, interpreted according to Cochrane guidelines (0–40%: might not be important; 30–60%: moderate heterogeneity; 50–90%: substantial heterogeneity; 75–100%: considerable heterogeneity). Given the presence of heterogeneity, a random-effects model was employed to accommodate between-study variability, recognizing that this approach may result in broader confidence intervals and more conservative estimates. All statistical analyses were performed using MetaXL version 5.3 (EpiGear International, Queensland, Australia), an add-on for Microsoft Excel specifically designed for meta-analyses of proportions.

## 3. Results

### 3.1. Study Selection

A total of 10 randomized controlled trials (RCTs), all published in English in medical journals indexed by Scopus and Web of Science, were included in the meta-analysis (Table 1). Appendix A lists the full texts excluded and the reasons for their exclusion.

Publication Bias

Overall, data were symmetrically distributed around the pooled estimates, suggesting no significant evidence of publication bias (Figure 2). However, mild asymmetry was noted for certain events, such as fever and headache, which may reflect underrepresentation of trials with low estimates and high precision. This finding should be interpreted with caution, as it may also be explained by the relatively small number of studies per subgroup.

### 3.2. General Profile of Adverse Events

Across the included studies, 18 distinct adverse events were reported, of which 4 were classified as local and 14 as systemic. Local adverse events included pain, erythema, swelling, and induration at the injection site. Systemic events included fatigue, headache, fever, arthralgia, myalgia, nausea, chills, loss of appetite, insomnia, persistent crying, vomiting, diarrhea, irritability, and rash (Table 1).

### 3.3. Local Adverse Events

Pain at the injection site was the most frequent local reaction, with a pooled incidence of 84%. Despite the consistency of findings across studies, statistical heterogeneity was significant (I^2^ = 92%). Less frequent local events included erythema (30%), swelling (25%), and induration (27%). These showed higher variability, with incidence ranging from 10% to 50% and significant heterogeneity in all cases

### 3.4. Systemic Adverse Events

Among systemic events, irritability was the most common, with a pooled incidence of 64% and substantial heterogeneity (I^2^ = 78%). Fatigue was reported in 51% of participants, followed by headache at 42%, both showing considerable heterogeneity (I^2^ = 87% and 85%, respectively). Moderate frequencies were observed for myalgia (40%), insomnia (40%, persistent crying (50%), and loss of appetite (27%). Levels of heterogeneity varied, with persistent crying and loss of appetite showing the highest variability. Less frequent systemic events included fever (8%), arthralgia (16%), nausea (19%), chills (20%), vomiting (11%), diarrhea (17%), and rash (7%). Fever showed extreme heterogeneity (I^2^ = 94%), partially explained by outlier results such as those reported by O’Connor et al. [19]). In contrast, arthralgia was among the few adverse events with non-significant heterogeneity. Most of the other, less frequent events showed low or negligible heterogeneity.

Additional details on local and systemic adverse event data are presented in Appendix A.

### 3.5. Heterogeneity and Statistical Model

The high variability observed across trials justified the use of a random-effects model for pooled analyses. Notably, heterogeneity was substantial for several adverse events: injection-site pain (92%), fatigue (87%), headache (85%), fever (94%), and irritability (78%), supporting the need to account for between-study differences beyond random error.

The analysis of adverse events revealed considerable variability across studies, both in terms of risk and frequency. As shown in Figure 3, the odds ratios (OR) and their 95% confidence intervals (95% CI) indicated heterogeneity in the association between treatment and specific adverse events, including headache, pain, erythema, edema, fatigue, fever, and irritability. OR values greater than 1 suggested an increased risk, whereas values below 1 indicated a reduced risk, with pooled estimates represented as diamonds summarizing the overall effect across studies. Additionally, Figure 4 illustrates the estimated proportions of each adverse event reported in the included studies, showing the frequency and distribution of both local and systemic reactions. The proportions varied by study, reflecting differences in reporting and population characteristics. Taken together, these findings provide a comprehensive overview of the safety profile, highlighting both the relative risk and the occurrence rates of adverse events.

### 3.6. Publication Bias

Overall, data were symmetrically distributed around the pooled estimates, suggesting no significant evidence of publication bias. However, mild asymmetry was noted for certain events, such as fever and headache, which may reflect underrepresentation of trials with low estimates and high precision. This finding should be interpreted with caution, as it may also be explained by the relatively small number of studies per subgroup.

## 4. Discussion

This meta-analysis synthesized the evidence on local and systemic adverse events associated with the administration of the 4CMenB vaccine. Injection-site pain emerged as the most frequent adverse event, with a pooled incidence of 84%, followed by systemic reactions such as irritability (64%), fatigue (51%), and persistent crying (50%). By contrast, fever (8%) and rash (7%) were infrequent. These results are consistent with individual randomized controlled trials (RCTs) and systematic assessments of MenB vaccines [17,18,19,21,22,23].

Local reactogenicity, particularly injection-site pain, was consistently higher for 4CMenB than for other pediatric vaccines, including hepatitis A, where pain ranged between 26% and 53% [18,21]. Gossger et al. [17] reported similarly high frequencies of injection-site pain in a large phase IIb/III RCT, confirming the findings of our pooled analysis. Subsequent studies with novel MenB-containing combinations such as MenABCWY have also described comparable rates of local reactogenicity, reinforcing the robustness of these results [21].

Systemic adverse events were more heterogeneous. Irritability, fatigue, and persistent crying were the most common systemic events, although study size and methodology influenced estimates. For instance, irritability was reported in smaller trials [4,20], whereas fatigue was consistently documented in larger RCTs [22,23,24]. Persistent crying, although distressing for parents, has been repeatedly confirmed as a transient adverse event in infants [24,25,26]. Other systemic events, such as headache, myalgia, and insomnia, were observed with moderate frequency (40–42%), while arthralgia, nausea, chills, loss of appetite, vomiting, diarrhea, and rash were reported less frequently (<27%) [25,26,27,28,29,30]. Importantly, most systemic events resolved spontaneously within 24–48 h, aligning with prior safety reviews of routine pediatric immunizations [26,27].

Fever showed a relatively low pooled incidence (8%), although heterogeneity was substantial. O’Connor et al. [19] reported higher fever rates in infants, while other trials demonstrated that fever was more common when 4CMenB was co-administered with routine vaccines [14,15,23,27]. The JAMA RCT by Vesikari et al. [27] similarly highlighted the increased risk of febrile reactions when MenB vaccination coincided with other infant immunizations. To mitigate this, prophylactic paracetamol and separate administration have been recommended by regulatory agencies such as the EMA [37] and the CDC [38], without compromising immunogenicity [24].

From a broader perspective, the balance between reactogenicity and protection must be considered. While adverse events such as pain and irritability are more frequent than with many other routine vaccines, they are generally mild, self-limited, and outweighed by the vaccine’s protective effect. The effectiveness of 4CMenB in reducing IMD has been demonstrated in national programs, such as the United Kingdom, where a reduced infant schedule achieved a 71% effectiveness against MenB disease [31]. This real-world evidence complements the immunogenicity and persistence data from multiple RCTs [22,23,28,29].

Heterogeneity was notable for only 5 of the 17 adverse events. It was substantial for several outcomes, including injection-site pain, fatigue, headache, fever, and irritability. Such heterogeneity may have influenced the reliability of the pooled estimates for these outcomes, and results should therefore be interpreted with caution.


**Strengths and Limitations**


A key strength of this meta-analysis is its deliberate focus on 4CMenB, chosen to preserve clinical and methodological comparability across studies. The two licensed MenB vaccines, 4CMenB (Bexsero) and rLP2086 (Trumenba), differ substantially in antigenic composition, dosing schedules, co-administration practices, and study populations, which would have introduced considerable heterogeneity and limited the robustness of pooled analyses. Moreover, 4CMenB is by far the most widely implemented MenB vaccine in infant immunization schedules worldwide, making its safety profile particularly relevant from a public health perspective. Additional strengths include strict adherence to PRISMA guidelines, the exclusive inclusion of phase II and III RCTs, and the use of a multilingual search strategy that enhanced external validity [16]. The comprehensive assessment of both local and systemic outcomes provides a more detailed and nuanced picture of the safety profile than individual trials alone.

This study also has important limitations. Substantial heterogeneity was observed for several outcomes, likely reflecting methodological differences in adverse event definitions, study populations, and follow-up periods. Some systemic events were reported in only a limited number of studies, reducing the precision of pooled estimates.

Furthermore, the limited number of pooled RCTs precluded subgroup analyses based on vaccination schedules, doses, gender, or age groups.

Although immune responses and tolerability may vary across age groups, many reports did not stratify adverse events by age, preventing a reliable subgroup analysis. While some studies suggest that infants may experience higher reactogenicity, particularly fever and irritability, compared with older age groups, the evidence remains inconsistent and insufficient to support robust comparative conclusions. Future research should aim to generate more granular, age-specific safety data to clarify potential differences across populations and to strengthen the evidence base for tailored vaccination strategies.


**Implications for Public Health**


Overall, the findings confirm that 4CMenB has an acceptable safety profile, consistent with prior trials and regulatory evaluations [17,18,19,21,22,23,24,25,26,27,28,29,30,37,38]. Most adverse events were mild, short-lived, and comparable to those of other childhood vaccines [26]. However, perceptions of reactogenicity can influence vaccine acceptance. Qualitative research has shown that parental concerns about fever and discomfort are among the most common barriers to MenB vaccination [34]. Transparent communication regarding the nature and frequency of adverse events, combined with reassurance about their transience and preventive strategies, is therefore essential to maintain confidence in MenB immunization programs.

Beyond safety, 4CMenB confers substantial public health benefits. A nationwide matched case–control study in Spain demonstrated that complete vaccination (at least two doses) was 76% effective against invasive meningococcal disease of any serogroup and 71% effective specifically against serogroup B disease in children under five years of age. Even partial vaccination (single dose) offered notable protection, with effectiveness reaching 64% against serogroup B [12]. Taken together, the evidence indicates that 4CMenB combines an acceptable and manageable safety profile with substantial effectiveness in preventing invasive meningococcal disease, underscoring a clearly favorable benefit–risk balance that supports its inclusion in routine immunization programs.

## 5. Conclusions

This meta-analysis confirms that 4CMenB is associated with a high frequency of local adverse events, particularly injection-site pain, and with common but mostly mild and transient systemic reactions such as irritability, fatigue, and persistent crying. Less frequent events, including fever and rash, were generally self-limited. Despite substantial heterogeneity across studies, the overall safety profile of 4CMenB remains acceptable and consistent with prior regulatory evaluations and real-world evidence. Taken together with its demonstrated protective effectiveness, these findings support the continued use of 4CMenB in immunization programs and underscore a favorable benefit–risk balance, while highlighting the importance of transparent communication to address parental concerns and sustain confidence in meningococcal B vaccination.

## Figures and Tables

**Figure 1 vaccines-13-01030-f001:**
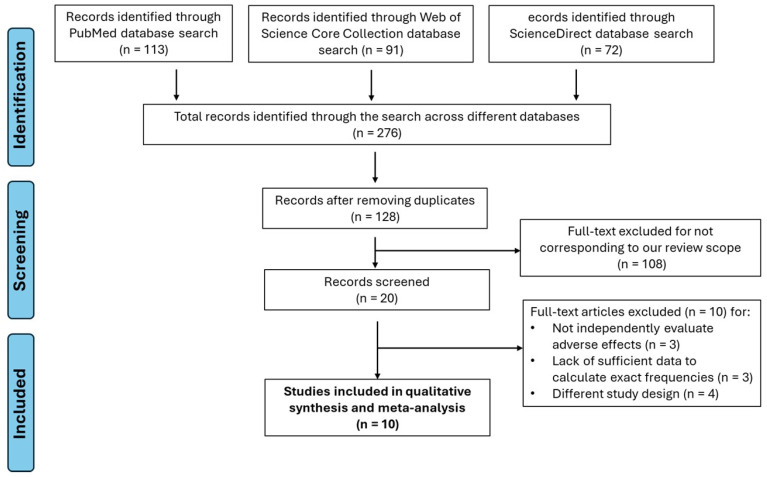
PRISMA flow diagram of the study selection process for the systematic review and meta-analysis. The diagram illustrates the phases of identification, screening, and final selection of studies included in the meta-analysis on adverse effects of the 4CMenB (Bexsero^®^) vaccine. A total of 276 records were initially identified through searches in three electronic databases. After removing duplicates and assessing inclusion and exclusion criteria, 10 studies were finally included in the quantitative synthesis.

**Figure 2 vaccines-13-01030-f002:**
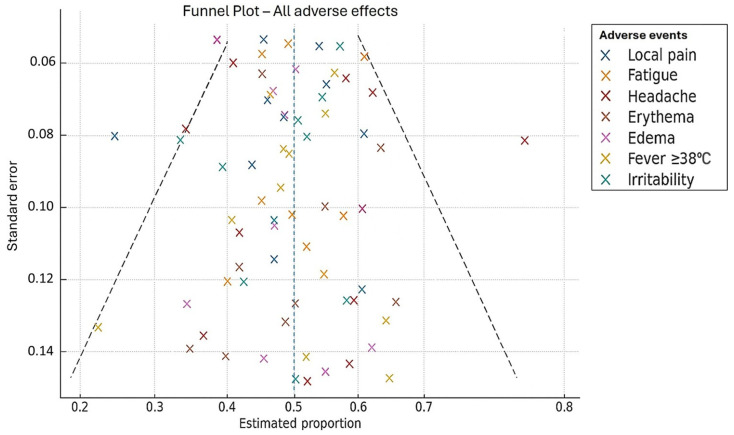
Funnel plot for the assessment of publication bias in adverse events following administration of the 4CMenB (Bexsero^®^) vaccine. The plot shows the distribution of standard errors against estimated proportions for the main adverse events reported after 4CMenB (Bexsero^®^) vaccination. Each point represents an individual study for a specific adverse event. The vertical dashed line indicates the overall mean proportion, and the funnel-shaped lines represent the 95% confidence limits in the absence of bias.

**Figure 3 vaccines-13-01030-f003:**
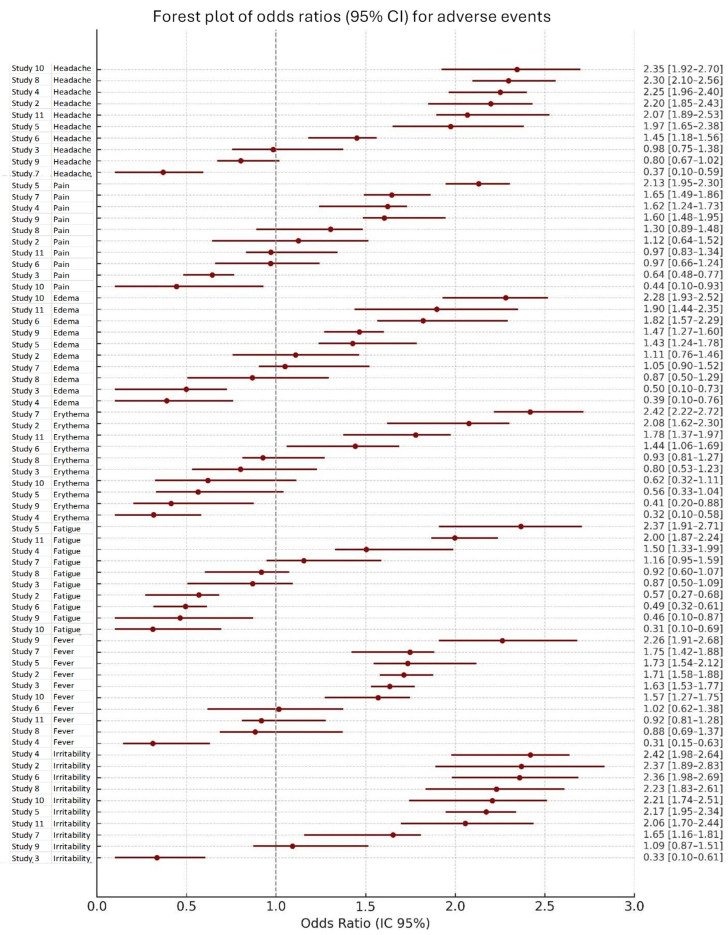
Forest plot—Odds ratios by adverse event. The plot presents the odds ratios (OR) and their 95% confidence intervals (95% CI) for adverse events reported in each study, showing the variability of effects across studies. The studies are organized by adverse events, including headache, pain, erythema, edema, fatigue, fever, and irritability. Each horizontal bar represents the confidence interval for the OR in each study, with the central point indicating the estimated OR. OR values greater than 1 indicate an increased risk of adverse events, whereas values below 1 indicate a reduced risk. The overall estimate for each adverse event is shown as a diamond, representing the pooled OR across all included studies.

**Figure 4 vaccines-13-01030-f004:**
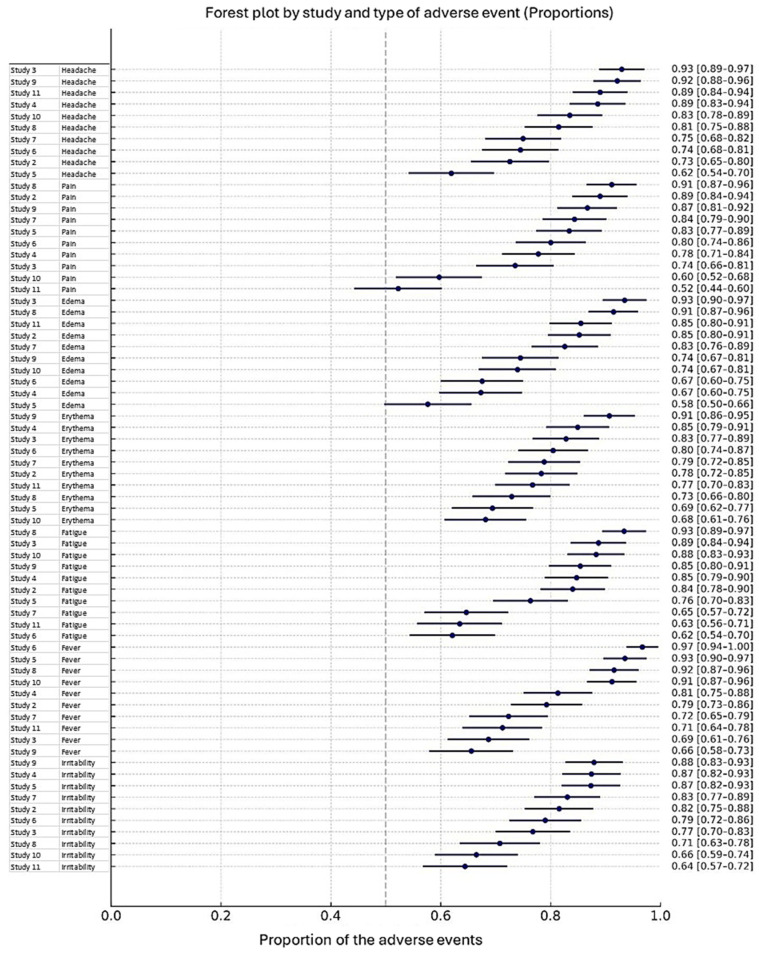
Forest plot by study and type of adverse event (Proportions). The plot shows the estimated proportions of each adverse event reported in each study, together with their 95% confidence intervals (95% CI). Each horizontal bar represents the confidence interval for the proportion of the adverse event in each study. The vertical dashed line marks the pooled proportion across studies. The studies are organized by type of adverse event, including headache, pain, erythema, edema, fatigue, fever, and irritability. The proportion values indicate the frequency of each adverse event among participants in the corresponding study.

**Table 1 vaccines-13-01030-t001:** Characteristics of the studies included in the meta-analysis and frequency of local and systemic adverse effects after administration of the 4CMenB (Bexsero^®^) vaccine.

Study ID/ Reference	Design	N	Age	Sex (M/F)	Local Adverse Effects (n, %)	Systemic Adverse Effects (n)
#2 [32]	Phase 2 CT	94	17.4 ± 4.64 years	40/54	Erythema: 10 (10.6%); Edema: 12 (12.8%); Induration: 12 (12.8%); Pain: 87 (92.6%)	Arthralgia (17), Fatigue (56), Nausea (17), Headache (39), Myalgia (38), Fever (1)
#3 [18]	Phase 2b CT	353	14.5 ± 3.1/16.8 ± 3.1 years	151/202	Erythema: 45 (12.7%); Edema: 0 (0.0%); Induration: 44 (12.5%); Pain: 328 (92.9%)	Chills (72), Arthralgia (40), Fatigue (190), Nausea (66), Headache (164), Myalgia (84), Fever (11), Loss of appetite (61)
#4 [19]	NR	92	8–12 weeks	NR	Erythema: 0 (0.0%); Edema: 0 (0.0%); Induration: 0 (0.0%); Pain: 92 (100.0%)	Fever (55)
#5 [33]	Phase 3b CT	254	20.9 ± 2.69 years	127/127	Erythema: 29 (11.4%); Edema: 43 (16.9%); Induration: 43 (16.9%); Pain: 250 (98.4%)	Arthralgia (63), Fatigue (140), Nausea (51), Headache (125), Myalgia (98), Fever (9)
#6 [4]	Phase 3b CT	228	NR	NR	Erythema: 102 (44.7%); Edema: 86 (37.7%); Induration: 102 (44.7%); Pain: 159 (69.7%)	Fever (54), Loss of appetite (47), Sleep disturbance (78), Persistent crying (132), Vomiting (28), Diarrhea (41), Irritability (121), Skin rash (28)
#7 [20]	Phase 3 CT	148	NR	NR	Erythema: 54 (36.5%); Edema: 34 (23.0%); Induration: 63 (42.6%); Pain: 75 (50.7%)	Fever (71), Loss of appetite (92), Sleep disturbance (79), Persistent crying (96), Vomiting (21), Diarrhea (28), Irritability (111), Skin rash (18)
#11 [34]	Phase 3	342	11–17	NR	Erythema: 156 (45.6%); Edema: 94 (27.5%); Induration: 93 (27.2%); Pain: 326 (95.3%)	Arthralgia (49), Fatigue (121), Nausea (64), Headache (117), Myalgia (192), Fever (9), Skin rash (14)
#13 [35]	Phase 2 CT	50	5 years	NR	Erythema: 47 (94.0%); Edema: 19 (38.0%); Induration: 42 (84.0%); Pain: 23 (46.0%)	Arthralgia (10), Fever (5), Loss of appetite (17), Sleep disturbance (18), Vomiting (5), Diarrhea (4), Headache (5), Irritability (24), Skin rash (2)
#15 [36]	Phase 2 CT	182	NR	NR	Erythema: 59 (32.4%); Edema: 32 (17.6%); Induration: 55 (30.2%); Pain: 63 (34.6%)	Fever (99), Loss of appetite (42), Sleep disturbance (66), Persistent crying (52), Vomiting (13), Diarrhea (31), Irritability (70), Skin rash (3)
#19 [17]	Phase 2b/3 CT	3330	11–17	NR	Pain: 2863 (86.0%)	Fatigue (1703), Headache (1412), Fever (123)

Summary of the clinical trials included in the meta-analysis, indicating the study design, sample size, and type of population. The most frequently reported local and systemic adverse effects after administration of the 4CMenB vaccine are presented, with the absolute number of cases and the relative percentage with respect to the total number of participants evaluated in each study. CT: Clinical trial; SD: Standard deviation; n: Number of participants with the event; N: Total number of participants evaluated; NR: not reported; %: Percentage with respect to the total evaluated in each study; Phase: Stage of the clinical trial development.

## Data Availability

Additional data are available in the Appendix A.

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
