# Peer review of "Safety Profile of the 4CMenB (Bexsero^®^) Vaccine: A Systematic Review and Meta-Analysis of Adverse Events in Clinical Trials"

_vaccines, 2025, doi:10.3390/vaccines13101030_

Round 1

Reviewer 1 Report

Comments and Suggestions for Authors

The meta-analysis manuscript entitled “Safety Profile of the 4CMenB (Bexsero®) Vaccine: A Systematic Review and Meta-Analysis of Adverse Events in Clinical Trials”, Meningococcal disease (MD) and Invasive meningococcal disease (IMD) remain a major public health concern worldwide. Therefore, assessing the quality of preventive measures developed is important for the combating of the disease. The authors estimate the incidence of local and systemic adverse events associated with the administration of the 4CMenB (Bexsero®) vaccine using a systematic review and meta data analysis methodologies. The manuscript reports several systemic adverse events but these events are below the benefits that the vaccine provides. This is an important additional to the vaccine field more especially in these times when we continue experiencing different aggresssive diseases. The manuscript is clear and follows the idea from the beginning to the end. The references used in the manuscript are relevant to the topic of discussions. The methodological design and the figures are clearly presented. There minor comments to the authors below.

Minor comments

Line 17: it is not clear what indicated mean in this sentence

Line 21: “was” should be were

Lines 22-23: “Human studies published in English, Spanish, French, German, and Italian were included” it is not clear what the authors mean by this sentence. How is this sentence differ from the previous sentence stating with “A systematic review…..)

The authors should carefully go through the manuscript to remove typos 

Author Response

Reviewer 1:

Comment 1: The meta-analysis manuscript entitled “Safety Profile of the 4CMenB (Bexsero®) Vaccine: A Systematic Review and Meta-Analysis of Adverse Events in Clinical Trials”, Meningococcal disease (MD) and Invasive meningococcal disease (IMD) remain a major public health concern worldwide. Therefore, assessing the quality of preventive measures developed is important for the combating of the disease. The authors estimate the incidence of local and systemic adverse events associated with the administration of the 4CMenB (Bexsero®) vaccine using a systematic review and meta data analysis methodologies. The manuscript reports several systemic adverse events but these events are below the benefits that the vaccine provides. This is an important additional to the vaccine field more especially in these times when we continue experiencing different aggressive diseases. The manuscript is clear and follows the idea from the beginning to the end. The references used in the manuscript are relevant to the topic of discussions. The methodological design and the figures are clearly presented.

Answer 1: Thank you.

Comment 2: Line 17: it is not clear what indicated mean in this sentence.

Answer 2: The word “indicated” was replaced by “recommended”.

Comment 3: Line 21: “was” should be “were”.

Answer 3: Sorry for this grammatical mistake. The word “was” was replaced by “were”.

Comment 4: Lines 22-23: “Human studies published in English, Spanish, French, German, and Italian were included” it is not clear what the authors mean by this sentence. How is this sentence differ from the previous sentence stating with “A systematic review…..).

Answer 4: We refer to our inclusion criteria. The sentence was corrected; Lines 22-23: “Human studies available in English, Spanish, French, German, or Italian were exclusively included”.   

Comment 5: The authors should carefully go through the manuscript to remove typos.

Answer 3: We regret the presence of typographical errors. The manuscript has since been thoroughly reviewed and all such errors have been corrected.

Reviewer 2 Report

Comments and Suggestions for Authors

Acknowledgment and Discussion of Methodological Limitations:

The authors claim that the overall risk of bias is low due to the inclusion of studies published in peer-reviewed journals. However, this statement is insufficient. It is crucial to acknowledge the lack of a formal risk-of-bias assessment using a validated tool such as the Cochrane RoB 2.0. This is a significant methodological limitation. The authors should include a paragraph in the "Methods" section that explicitly states this limitation and discusses its potential implications for the results.

Clarity on Dosing and Vaccination Schedules:

The clinical trials included in the review likely used different vaccination schedules and doses, particularly for different age groups. These variations could have an impact on the reactogenicity of the vaccine. The "Discussion" section could be strengthened by briefly addressing how these different schedules and doses might influence the reported adverse event rates.

Consideration of Demographic Factors:

The study does not differentiate between infants, children, and adults, nor does it address factors such as gender, race, or nationality. While the results reflect the overall safety of the vaccine across a broad population, this lack of stratified analysis may overlook potential differences among subgroups, which could reduce the precision and relevance of the findings. Furthermore, the paper does not provide information on the sample size distribution across different racial or national groups. If certain groups are well-represented, it may be worth analyzing the relationship between adverse events and these demographic factors.

Handling of Heterogeneity:

In several analyses, the I² statistic indicates a high level of heterogeneity (>75%) across studies, suggesting significant variability in results. However, the "Discussion" section only provides a brief mention of the potential impact of heterogeneity on the results, without offering a deeper analysis. A more thorough exploration of this issue could improve the robustness of the conclusions.

Clinical Considerations and Recommendations:

The "Discussion" section could benefit from further clinical insights, particularly regarding the selection of appropriate vaccination schedules for different populations. Additionally, recommendations for the clinical management of common adverse reactions, such as fever or injection site pain, would be valuable.

Justification for Using Random Effects Model:

The paper mentions the use of a random effects model to account for the expected heterogeneity across studies. It would strengthen the analysis to compare this model with other statistical approaches and provide additional justification for its selection as the most suitable method for this meta-analysis.

Author Response

Comment 1: Acknowledgment and Discussion of Methodological Limitations:

The authors claim that the overall risk of bias is low due to the inclusion of studies published in peer-reviewed journals. However, this statement is insufficient. It is crucial to acknowledge the lack of a formal risk-of-bias assessment using a validated tool such as the Cochrane RoB 2.0. This is a significant methodological limitation. The authors should include a paragraph in the "Methods" section that explicitly states this limitation and discusses its potential implications for the results.

Answer 1: The Cochrane RoB 2.0 tool was developed to assess the risk of bias in RCTs, aiming to enhance consistency, transparency, and methodological rigor in evaluating study quality.

A key feature of RoB 2.0 is its outcome-specific focus: bias is assessed for each individual outcome rather than for the study as a whole. In our meta-analysis, we identified only 10 RCTs that met the inclusion criteria and were published in Scopus- and WoS-indexed medical journals (added to the results’ section). These studies reported a total of 17 distinct adverse events, requiring 17 separate RoB 2.0 assessments for just 10 trials. Additionally, RoB 2.0 includes domains related to bias due to missing outcome data, bias in outcome measurement, and bias in the selection of reported results. However, these domains were not applicable in our analysis, given the strict inclusion and exclusion criteria and our focus on estimating the incidence of adverse events with their corresponding 95% confidence intervals. Finally, RoB 2.0 is subject to interpretive variability. Despite its structured guidance, signaling questions may be understood differently by reviewers, potentially leading to inconsistent bias assessments across studies.

Comment 2: Clarity on Dosing and Vaccination Schedules:

The clinical trials included in the review likely used different vaccination schedules and doses, particularly for different age groups. These variations could have an impact on the reactogenicity of the vaccine. The "Discussion" section could be strengthened by briefly addressing how these different schedules and doses might influence the reported adverse event rates.

Answer 2: Thank you for this important comment and observation. Only 10 published RCTs met our inclusion and exclusion criteria, allowing us to pool data on 17 adverse events. Due to the limited number of RCTs, subgroup analyses based on vaccination schedules, doses, or age groups were not feasible. This represents an additional limitation of our meta-analysis and has been acknowledged in the limitations section.

 Comment 3: Consideration of Demographic Factors:

The study does not differentiate between infants, children, and adults, nor does it address factors such as gender, race, or nationality. While the results reflect the overall safety of the vaccine across a broad population, this lack of stratified analysis may overlook potential differences among subgroups, which could reduce the precision and relevance of the findings. Furthermore, the paper does not provide information on the sample size distribution across different racial or national groups. If certain groups are well-represented, it may be worth analyzing the relationship between adverse events and these demographic factors.

Answer 3: Thank you for your comment. As noted in our previous response, the limited number of published RCTs meeting our inclusion and exclusion criteria did not allow for subgroup analyses based on factors such as gender, race, or nationality.

Comment 4: Handling of Heterogeneity:

In several analyses, the I² statistic indicates a high level of heterogeneity (>75%) across studies, suggesting significant variability in results. However, the "Discussion" section only provides a brief mention of the potential impact of heterogeneity on the results, without offering a deeper analysis. A more thorough exploration of this issue could improve the robustness of the conclusions.

Answer 4: Thank you for your comment and suggestion. We added a new paragraph jst before Strengths and Limitations section:

Heterogeneity was notable for only 5 of the 17 adverse events. It was substantial for several outcomes, including injection-site pain, fatigue, headache, fever, and irritability. Such heterogeneity may have influenced the reliability of the pooled estimates for these outcomes, and results should therefore be interpreted with caution.

Comment 5: Clinical Considerations and Recommendations:

The "Discussion" section could benefit from further clinical insights, particularly regarding the selection of appropriate vaccination schedules for different populations. Additionally, recommendations for the clinical management of common adverse reactions, such as fever or injection site pain, would be valuable.

Answer 5: Thank you for your comment and suggestion. We added several sentences to the discussion and recommendations sections addressing both the benefits and adverse events of the vaccine (in Red).

Comment 6: Justification for Using Random Effects Model:

The paper mentions the use of a random effects model to account for the expected heterogeneity across studies. It would strengthen the analysis to compare this model with other statistical approaches and provide additional justification for its selection as the most suitable method for this meta-analysis.

Answer 5: Thank you for your comment and suggestion. We added a new paragraph in Data synthesis and statistical analysis:

Given the presence of heterogeneity, a random-effects model was employed to accommodate between-study variability, recognizing that this method may produce broader confidence intervals and more cautious estimates.

Reviewer 3 Report

Comments and Suggestions for Authors

This manuscript is a resubmitted version reporting a meta-amalysis concerning AEs of the 4CMenB vaccine which is recommended for vaccination programs for newborn children from 2 months of age onward. Despite several weaknesses which are adequately described in the report, suitable results concerning AEs can be extracted, and is of potential interest to the readership of Vaccines:

Some amendments are recommended:

1. The authors describe in 2.4. Comparators "Although some trials included comparisons with placebo or other vaccines, the present meta-analysis focused exclusively on the frequency of adverse events after 4CMenB vaccination; therefore, direct group comparisons were not extracted." These studies need to be described with more details and a brief discussion.

In fact, the lack of a true control group, i.e. placebo-injected healthy participants, as well as a longer time interval to monitor AEs are frequent major shortcomings in vaccine trials. Only the comparison with a proper placebo group can provide data on the actual efficacy and safety of vaccines. The benefit can be ultimately quantified as an increased lifespan (or at least in an overall reduction in pathological conditions) of the vaccinated participant compared with the non-vaccinated one, while data of differently vaccinated particpants may obscure negative effects. 

2. The authors only refer to conventional approaches with problematic pharmaceuticals alike paracetamol for treatment. However, there are effective, low-risk therapies - such as the application of high dose ascorbic acid or chlorine dioxide water to combat bacterial/viral infections, which are worth to be mentioned as well.

Author Response

Reviewer 3:

Comment 1: This manuscript is a resubmitted version reporting a meta-analysis concerning AEs of the 4CMenB vaccine which is recommended for vaccination programs for newborn children from 2 months of age onward. Despite several weaknesses which are adequately described in the report, suitable results concerning AEs can be extracted, and is of potential interest to the readership of Vaccines.

Answer 1: Thank you.

Comment 2: The authors describe in 2.4. Comparators "Although some trials included comparisons with placebo or other vaccines, the present meta-analysis focused exclusively on the frequency of adverse events after 4CMenB vaccination; therefore, direct group comparisons were not extracted." These studies need to be described with more details and a brief discussion.

In fact, the lack of a true control group, i.e. placebo-injected healthy participants, as well as a longer time interval to monitor AEs are frequent major shortcomings in vaccine trials. Only the comparison with a proper placebo group can provide data on the actual efficacy and safety of vaccines. The benefit can be ultimately quantified as an increased lifespan (or at least in an overall reduction in pathological conditions) of the vaccinated participant compared with the non-vaccinated one, while data of differently vaccinated participants may obscure negative effects. 

Answer 2: Table 1 summarizes all RCTs that met the inclusion and exclusion criteria. The effectiveness of the 4CMenB vaccine has been previously reported in several studies and consensus documents (see references 10–13).

Although some of the included RCTs featured placebo or comparator vaccine arms, the primary focus of this meta-analysis was to evaluate the frequency of adverse events following 4CMenB vaccination. Accordingly, direct comparisons with control groups were not necessary, as the main objective was to generate a comprehensive estimate of adverse event incidence among individuals receiving 4CMenB. This approach is appropriate for characterizing the vaccine’s safety profile independently of relative comparisons and enables the pooling of data across all eligible studies.

Comment 3: The authors only refer to conventional approaches with problematic pharmaceuticals alike paracetamol for treatment. However, there are effective, low-risk therapies - such as the application of high dose ascorbic acid or chlorine dioxide water to combat bacterial/viral infections, which are worth to be mentioned as well.

Answer 3: Our meta-analysis did not explore the pharmaceutical or non-pharmaceutical management of AEs. We limited our commentary to the prophylactic use of paracetamol for mitigating specific AEs such as fever. The choice of paracetamol was based on recommendations from regulatory agencies, including the EMA and the CDC.

The JAMA RCT by Vesikari et al. [33] similarly highlighted the increased risk of febrile reactions when MenB vaccination coincided with other infant immunizations. To mitigate this, prophylactic paracetamol and separate administration have been recommended by regulatory agencies such as the EMA [34] and the CDC [35], without compromising immunogenicity [25].